# Safety and Efficacy of Intermittent High-Dose Liposomal Amphotericin B Antifungal Prophylaxis in Haemato-Oncology: An Eight-Year Single-Centre Experience and Review of the Literature

**DOI:** 10.3390/jof6040385

**Published:** 2020-12-21

**Authors:** Jonathan Youngs, Jen Mae Low, Laura Whitney, Clare Logan, Janice Chase, Ting Yau, Matthias Klammer, Mickey Koh, Tihana Bicanic

**Affiliations:** 1Institute of Infection & Immunity, St George’s University of London, Cranmer Terrace, Tooting, London SW17 0RE, UK; clogan@sgul.ac.uk; 2Department of Infection, St George’s University Hospitals NHS Foundation Trust, Blackshaw Rd, Tooting, London SW17 0QT, UK; 3Clinical Academic Group in Infection and Immunity, St George’s University of London, Cranmer Terrace, London SW17 0RE, UK; mickey.koh@stgeorges.nhs.uk; 4St George’s University Hospitals NHS Foundation Trust, Blackshaw Rd, Tooting, London SW17 0QT, UK; jen.low@nhs.net; 5Pharmacy Department, St George’s University Hospitals NHS Foundation Trust, Blackshaw Rd, Tooting, London SW17 0QT, UK; lauracwhitney@yahoo.co.uk (L.W.); janice.chase@nhs.net (J.C.); ting.yau@nhs.net (T.Y.); 6Department of Haematology, St George’s University Hospitals NHS Foundation Trust, Blackshaw Rd, Tooting, London SW17 0QT, UK; matthias.klammer@stgeorges.nhs.uk

**Keywords:** hematopoietic stem-cell transplantation (HSCT), acute leukaemia, graft-versus-host disease (GvHD), antifungal prophylaxis, amphotericin B, triazoles

## Abstract

Triazoles remain first-line agents for antifungal prophylaxis in high-risk haemato-oncology patients, but their use is increasingly contraindicated due to drug–drug interactions and additive toxicities with novel treatments. In this retrospective, single-centre, observational study, we present our eight-year experience of antifungal prophylaxis using intermittent high-dose liposomal Amphotericin B (L-AmB). All adults identified through our Antifungal Stewardship Programme as receiving L-AmB prophylaxis at 7.5 mg/kg once-weekly between February 2012 and January 2020 were included. Adverse reactions, including infusion reactions, electrolyte loss, and nephrotoxicity, were recorded. ‘Breakthrough’ invasive fungal infection (IFI) occurring within four weeks of L-AmB was classified using European Organization for Research and Treatment of Cancer/Invasive Fungal Infections Cooperative Group and the National Institute of Allergy and Infectious Diseases Mycoses Study Group (EORTC/MSG) criteria. Moreover, 114 courses of intermittent high-dose L-AmB prophylaxis administered to 92 unique patients were analysed. Hypokalaemia was the most common grade 3–4 adverse event, with 26 (23%) courses. Grade 3 nephrotoxicity occurred in 8 (7%) and reversed in all six patients surviving to 90 days. There were two (1.8%) episodes of breakthrough IFI, one ‘probable’ and one ‘possible’. In this study, the largest evaluation of intermittent high-dose L-AmB prophylaxis conducted to date, toxicity was manageable and reversible and breakthrough IFI was rare. L-AmB prophylaxis represents a viable alternative for patients with a contraindication to triazoles.

## 1. Introduction

Antifungal prophylaxis is recommended for selected patients undergoing cytotoxic chemotherapy and hematopoietic stem-cell transplantation (HSCT) to reduce the risk of invasive fungal infection (IFI) including candidiasis, aspergillosis and mucormycosis [1]. Because of their broad antifungal spectrum, ease of administration, and ability to achieve target serum concentrations, mould-active triazoles (e.g., itraconazole/posaconazole) remain first line agents in high-risk patients such as those with acute myeloid leukaemia/myelodysplastic syndromes (AML/MDS) or undergoing treatment for graft-versus-host disease (GvHD).

Triazole use is, however, often contraindicated due to their hepatic metabolism and toxicity as well as QTc prolongation. Itraconazole and posaconazole are potent inhibitors of cytochrome P450 3A4 and p-glycoprotein, so numerous drug–drug interactions must also be considered [2]. Alongside commonly used drugs a variety of interactions and additive toxicities exist between triazoles and both established and novel treatments for hematological malignancy (Appendix A) [3,4,5,6].

Conventional Amphotericin B deoxycholate (C-AmB) is the most broad-spectrum antifungal agent, with crucial activity against most *Aspergillus* spp., *Candida* spp., *Mucor* spp., and other environmental moulds. Its unfavourable toxicity profile, notably electrolyte abnormalities, nephrotoxicity, and anaemia, has hindered its use as a prophylactic agent [2,7,8]. AmBisome^®^ (L-AmB) is a liposomal formulation of Amphotericin B with a more favorable toxicity profile [9,10]. In an RCT of pyrexic neutropenic adults, L-AmB was as effective as C-AmB, but associated with 2–6 times fewer drug-related side effects and severe drug-related side effects were seen in only 1% vs. 12% (*p* < 0.01) [10].

To treat IFI, L-AmB is administered daily (dose 3–5 mg/kg/day) which, given its intravenous route, is impractical for outpatients requiring antifungal prophylaxis. However, when administered at higher doses (7.5–15 mg/kg), therapeutic levels of L-AmB persist in tissues for over a week without an associated increase in toxicity, supporting intermittent high dosing [2,11,12,13,14]. Administering L-AmB at 7.5 mg/kg (once-weekly) and 15 mg/kg (once only) to myeloma patients undergoing allograft transplantation resulted in mean serum and buccal concentrations of the drug well in excess of *Candida* and *Aspergillus* MICs after seven days [12] and doses up to 7.5–15mg/kg/day were well tolerated in a maximally-tolerated dose pharmacokinetic (PK) study, although severe hypokalaemia was more common in patients receiving >10 mg/kg/day (0% vs. 35%, *p* = 0.006) [15].

Since 1999, 15 studies have reported on the safety and efficacy of different regimens of intravenous L-AmB prophylaxis in haemato-oncology patients: 12 were prospective and 5 RCTs, of which only one (2017) trial had >100 patients per arm (Table 1) [9,11,12,14,16,17,18,19,20,21,22,23,24,25,26,27]. All eight cohorts using once-weekly doses ≥7.5mg/kg have been small (<50 patients). In this single-centre, retrospective observational study, we report our eight-year experience of the safety (toxicity/adverse events) and efficacy (incidence of breakthrough IFI) of L-AmB prophylaxis in high risk haemato-oncology patients encompassing the largest cohort of high-dose, intermittent L-AmB prophylaxis reported to date.

## 2. Materials and Methods

St George’s hospital is a 1300-bed London tertiary referral centre treating patients with haematological malignancy. It performs autologous and allogeneic HSCT; 29/year in 2009 rising to an average of 50/year currently. Since October 2010, an Antifungal Stewardship (AFS) programme has operated, incorporating weekly review of antifungal prescriptions for all adult inpatients and Haematology Day Care Unit attendees.

Over the study period, our local antifungal policy recommended itraconazole and posaconazole as 1st and 2nd line prophylactic agents for patients at high risk of IFI [1], who are routinely cared for in positive-pressure HEPA-filtered single rooms. Where triazoles were contraindicated, not tolerated, or deemed unsuitable, L-AmB was used as a 3rd line agent. Before February 2011, L-AmB was administered at 1mg/kg/day. Between February 2011–February 2012, the policy changed to 3 mg/kg × 3/week. In February 2012, following a positive report from another London institution [28] and literature review of intermittent high-dose L-AmB prophylaxis, a regimen of 7.5 mg/kg once-weekly was instituted and continues to the present day. Where antifungal prophylaxis relates to intensive chemotherapy (or a novel agent), it is usually commenced at the time of, or prior to, the first dose.

Our local policy is that after a 1 mg test dose, L-AmB is infused over 2 h without routine pre-medication in the first instance. In the event of a reaction, the infusion rate is slowed and medications (e.g., chloramphenamine, paracetamol, and hydrocortisone) are administered at the discretion of the treating physician for both the current and future infusions.

At-risk patients (e.g., all those receiving antifungal prophylaxis) undergo investigation for IFI if there is persistent fever (>72 h) despite broad-spectrum antibiotics accompanied by suggestive symptoms or signs. This investigation comprises blood cultures, a CT chest (plus CT sinus/head/abdomen as indicated), serum galactomannan and serum beta-D-glucan. A bronchoalveolar lavage (BAL, with prolonged fungal culture, galactomananan and *Pneumocystis* pneumonia polymerase chain reaction testing) is pursued in response to CT chest findings. Whilst these investigations are ongoing, prophylaxis is switched to empirical pre-emptive treatment with L-AmB (3 mg/kg/day) if the patient is unwell. Voriconazole is initiated if investigations suggest invasive aspergillosis. Asymptomatic patients are not routinely screened for IFI.

As part of ongoing clinical service evaluation, records of AFS reviews, including patient demographics, antifungal drug/indication and recommendations made, were prospectively entered into secure Microsoft Excel (v2003) and subsequently REDCap databases. We conducted a retrospective evaluation of L-AmB prophylaxis in haemato-oncology patients, extracting data on any patient receiving ‘L-AmB’ for the indication ‘prophylaxis’ between October 2010–January 2020 but only patients receiving intermittent high-dose L-AmB (7.5 mg/kg once weekly) were included in the main analysis. Patients without an underlying haematological condition, primarily under the care of another medical institution or for whom the indication for L-AmB was treatment rather than prophylaxis, were also excluded.

Information extracted included: underlying haematological condition (including HSCT); rationale for antifungal prophylaxis; reason triazole contraindicated; L-AmB dosing/duration and reason for ceasing L-AmB. Missing data were retrieved from paper and electronic medical records and pharmacy, laboratory, and microbiology databases.

Adverse reactions to L-AmB were recorded including nephrotoxicity and hypokalaemia or hypomagnesaemia related to L-AmB administration. Toxicity was graded as per the Common Terminology Criteria for Adverse Events [29].

Episodes of ‘breakthrough’ IFI were classified as ‘proven’, ‘probable’ or ‘possible’ based on EORTC/MSG criteria [30]. In line with a consensus statement [31] we took the date of breakthrough IFI as the first radiological/clinical sign/mycological finding within the period of antifungal prophylaxis. This in turn depends on antifungal drug half-life and dosing schedule: we defined L-AmB prophylaxis course length as extending until the next dose would be due (i.e., one week with once-weekly prophylaxis) but included any IFI within 4 weeks of the last dose of L-AmB as a breakthrough, based on the above PK data.

Data were analysed using Microsoft Excel v2019 and GraphPad Prism v8.4.2 (GraphPad software, LLC, San Diego, CA, USA), using Fischer’s exact test for categorical variables and Mann-Whitney U test for continuous variables.

After utilising the NHS Health Research Authority Decision Tools [32], no formal ethics approval was considered necessary as persons accessing patient-identifiable data were members of the direct care team doing so as part of a clinical service evaluation.

## 3. Results

During the nine-year period from October 2010 to January 2020, 147 L-AmB prophylaxis courses were identified. Thirty-three were excluded: low-dose courses (daily 1 mg/kg/d or thrice weekly 3 mg/kg, *n* = 24), treatment not prophylaxis (*n* = 5); patient primarily under another institution (*n* = 2); no underlying hematological condition (*n* = 2). The remaining 114 courses of intermittent high-dose (7.5 mg/kg weekly) L-AmB prophylaxis administered to 92 unique patients were analysed (all administered after February 2012). These courses amounted to 520 L-AmB infusions and 3640 patient-days of antifungal prophylaxis.

Baseline characteristics and rationale for L-AmB prophylaxis are shown in Table 2. Median (IQR) patient age was 42 (29–57) years and 57% were male. The median course length was 24 (7–147) days deriving from 3 (1–21) infusions. The most common indication for antifungal prophylaxis was intensive chemotherapy/novel agent for induction/relapse (77%), the majority (78%) of whom had acute leukaemia. Fourteen patients (12%) received antifungal prophylaxis in relation to HSCT and ten (9%) due to GvHD. The most frequent contra-indications to a triazole were to avoid a drug–drug interaction (51%) or abnormal liver-function tests (39%). Drug–drug interactions were usually related to vincristine or myelotarg (88%).

Of the 88 patients receiving antifungal prophylaxis in relation to intensive chemotherapy/novel agent, 79 (90%) were severely neutropenic (<0.5 × 10^9^/L) at some point during their L-AmB course. Severe neutropenia lasted over 10 days in 56 (71%) of these courses and occurred for a median of three days (IQR, 0–7) after L-AmB commenced.

Adverse events (AEs) and incidence of breakthrough IFI are summarised in Table 3. Nephrotoxicity and electrolyte loss were the most frequent AEs. The most common grade 3–4 AE was hypokalaemia; occurring in 26 (23%) of L-AmB courses. Hypomagnesemia was common (66%) but only one instance was grade 3–4. Grade 3 nephrotoxicity was associated with eight (7%) L-AmB courses, with no instances of grade 4 nephrotoxicity. Of these eight patients, seven had acute leukaemia and the rationale for prophylaxis was either allograft (*n* = 3), intensive chemotherapy (*n* = 4) or GvHD (*n* = 1) so all were likely to be receiving concomitant nephrotoxic medications. Excluding two patients who died <30 days, creatinine returned to baseline in the remaining six by 90 days (3 within30 days) [33]. All infusion reactions were mild but 3 of 5 acute allergic reactions were grade 3–4. L-AmB prophylaxis was discontinued due to toxicity in 17 (15%) of courses; infusion reaction (*n* = 7), acute allergic reaction (*n* = 5) and nephrotoxicity (*n* = 5) (Table 2).

In 114 courses of high dose intermittent L-AmB prophylaxis there were just 2 (1.8%) episodes of breakthrough IFI. This incidence remained low (2.4%) even if only patients conventionally considered high-risk are considered, e.g., intense chemotherapy for AML/MDS (*n* = 24), treatment for GvHD (*n* = 10), and <100 days post allograft (*n* = 7). One patient undergoing intensive chemotherapy for HLH developed ‘probable’ IFI 7 days into L-AmB prophylaxis [30]. Chest imaging revealed new nodular perihilar air space opacities and *Aspergillus fumigatus* was isolated twice from sputum culture. The second patient had a background of sibling allograft for ALL and developed ‘possible’ IFI two weeks into L-AmB prophylaxis for immunosuppression related to GvHD. CT chest revealed bilateral peribronchovascular consolidation with areas of discrete nodules and surrounding ground-glass change supportive of fungal infection. Sputum culture and serum galactomannan were negative, but serum beta-D-glucan was raised at 469 pg/mL. Both patients died within 90-days whereas the 90-day mortality for those without IFI was 8/112 (7%) (Table 3).

A sensitivity analysis was performed comparing intermittent high-dose L-AmB to the historic cohort of 24 patients that received low-dose AmB prophylaxis between 2010–11. There were no significant differences in nephrotoxicity (any grade, 57% vs. 42%, *p* = 0.18) (grade 3–4, 7% vs. 8%, *p* = 0.69), hypokalaemia (any grade, 66% vs. 79%, *p* = 0.24) (grade 3–4, 23% vs. 37.5%, *p* = 0.1), hypomagnesaemia (66% vs. 88%, *p*=0.05) or proven/probable IFI (1% vs. 4%, *p* = 0.3).

## 4. Discussion

Whilst novel targeted therapies represent an exciting development in the treatment of haematological malignancy they present new challenges for antifungal prophylaxis [5]. Drugs such as inotuzumab ozogamicin, ibrutinib, FLT3 inhibitors and venetoclax all have significant interactions/additive toxicities with triazoles and some, such as ibrutinib, are associated with a high incidence of IFI [34,35]. Safe, effective oral alternatives to azoles are urgently needed, with several promising novel agents in clinical development [36,37,38,39,40]. In the meantime, intravenous agents remain the most attractive option (although nebulised L-AmB may be an alternative [41]). Whilst echinocandins are an option, they have a narrower spectrum than L-AmB, require daily intravenous dosing, and are fungistatic (rather than -cidal) against Aspergillus spp. [2]. Our eight-year experience of prophylactic L-AmB in high risk haemato-oncology patients includes the largest series of intermittent high-dose courses to date, given for a median of 3–4 weeks. We found that these prophylactic regimens were well tolerated with reversible toxicities and associated with a low (1.8%) rate of breakthrough IFI.

The pharmacokinetics of L-AmB support intermittent high-dose administration for several reasons. Firstly, the efficacy of L-AmB is concentration-dependent so achievement of a high maximum drug concentration in plasma (Cmax) compared to the fungal MIC—the Cmax/MIC ratio, is desirable [2,11,12]. Secondly, L-AmB (more so than C-AmB) accumulates in the reticulo-endothelial system and other tissues for several weeks after administration, in a dose-dependent manner [13]. Lastly, L-AmB exhibits non-linear pharmacokinetics meaning that it is preferentially cleared at higher doses [14].

These characteristics mean that when administered at higher doses, therapeutic levels of L-AmB persist in tissues without an associated increase in toxicity. In one study, levels of L-AmB above the MIC of most fungi were maintained for one week in mice lungs and six weeks in kidney/spleen after administration of 15 mg/kg × 3/week for 2–5 weeks. Drug levels in kidney/spleen 3–6 weeks post treatment were similar whether a daily (2.4 mg/kg × 5/week) or intermittent (15 mg/kg once-weekly) dose was used [42]. In another study, a single high-dose of L-AmB protected immunosuppressed mice against challenge with *Histoplasma spp.* seven days later and survival increased with higher dosing: 20% (5 mg/kg) to 60% (10mg/kg) and 80% (20 mg/kg) [43].

The largest clinical trial of standard-dose L-AmB prophylaxis to date has been the AmbiGuard study (*n* = 355), a multi-centre randomized placebo-controlled trial of L-AmB prophylaxis (5 mg/kg twice-weekly) in induction chemotherapy for ALL (Table 1) [21]. There were more serious AEs considered related to the study drug with L-AmB compared to placebo (8% vs. 2%, *p* = 0.02) as well as more hypokalaemia (any grade—35% vs. 18%, *p* < 0.001) and creatinine elevation (any grade—9% vs. 0%, *p* < 0.001). Interestingly, however, patients receiving placebo were as likely to have their study drug stopped due to an AE as those receiving L-AmB (22% vs. 27%, *p* = 0.37), suggesting that some of the toxicity observed with L-AmB may be due to concurrent chemotherapy/HSCT conditioning.

In our study, as in others, the most common L-AmB toxicities were renal. Electrolyte loss and nephrotoxicity occurred in 66% and 57% of patients respectively. Grade 3 nephrotoxicity occurred in 7%, possibly exacerbated by concomitant use of other nephrotoxic agents, and was reversible in all survivors by 90 days. Grade 3–4 hypokalaemia occurred in 23% of patients, a figure between that of the placebo and L-AmB arms of AmbiGuard (18% and 35%, *p* < 0.001) (Table 3). Careful monitoring and pre-emptive potassium and magnesium replacement can mitigate the risk of this predictable AmB-related AE [44]. In patients with deranged liver function tests the risk of reversible renal toxicity with L-AmB must be weighed against the risk of drug induced liver injury with triazoles. Our policy is to continue with triazoles in patients with cholestatic liver enzyme derangement (alkaline phosphatase and gamma glutamyltransferase) and modest elevations in transaminases (alanine transaminase <150 IU/L), with exclusion of other causes and close laboratory monitoring for worsening transaminitis

In a sensitivity analysis we found no suggestion of increased AEs associated with intermittent high-dose L-AmB compared to the 24 patients that received lower doses, a finding echoed in other studies. In a retrospective analysis of adults with newly diagnosed AML/high-risk MDS, drug-related side effects were similar between patients receiving L-AmB prophylaxis 3 mg/kg × 3/week and 9 mg/kg once-weekly (14% vs. 12%) [26]. In one study that employed L-AmB at 15 mg/kg, there were mild infusion-related reactions in only six out of 53 (11%) of infusions and only one led to cessation [16]. The authors postulated their long infusion time of 6 h reduced AEs (our local policy is to administer over 2 h).

Our 15% rate of discontinuation due to toxicity is lower than that seen in a prospective study using the same L-AmB dose (33.3%) [24] and AmbiGuard (27%) [21]. This may reflect the patient popuations involved or protocol-driven discontinuationin a clinical trial. In the PROPHYSOME study [14] the trial group suspended enrolment into the SCT arm after AEs led to four of eight patients discontinuing L-AmB (10 mg/kg once-weekly) after the first dose [14]. All patients received at least one concomitant nephrotoxic medication and all eight SCT patients received cyclosporin. Interestingly, L-AmB was better tolerated in patients receiving chemotherapy for acute leukaemia, which represents most (76%) of our intermittent high-dose cohort. We too found suggestion that L-AmB toxicity prompting discontinuation may be more common with HSCT than with intensive chemotherapy, although the difference was not statistically significant, with four out of 14 (29%) vs. 12 out of 87 (14%) (*p* = 0.2). HSCT patients are frequently on concomitant nephrotoxic agents such as ciclosporin, acyclovir and aminoglycosides.

In our cohort, only two (1.8%) of 114 L-AmB courses were associated with breakthrough IFI, of which only one was proven/probable. This rate remained low (2.4%) when including only patients conventionally considered high-risk for IFI. Studies on L-AmB prophylaxis report IFI occurring over varying time periods ranging between only during prophylaxis to up to a year after the last dose (Table 1). This may partly explain why our incidence of proven/probable IFI is lower than the 5–10% reported [11,21,24] elsewhere.

Prospective comparative data on the efficacy of intermittent high-dose L-AmB prophylaxis are scarce. In a prospective study by El-Cheikh et al., (2007) 21 adults receiving high-dose steroids for GvHD post-allograft transplantation were administered L-AmB prophylaxis 7.5 mg/kg once-weekly. Only one (5%) instance of proven/probable IFI was observed, two months after discontinuation of L-AmB prophylaxis [24]. In a subsequent retrospective study, the rate of proven/probable IFD was four out of 42 (10%) in the L-AmB group vs. 13 out of 83 (16%) in a comparator group of patients that received ‘other antifungal prophylaxis’ (a triazole in 87%) [11].

In terms of regular standard-dose L-AmB prophylaxis, in AmbiGuard (5 mg/kg twice-weekly) [21], there were fewer episodes of proven/probable IFI with L-AmB compared to placebo, but the difference was not statistically significant: 18 in 228 (7.9%) vs. 13 in 111 (11.7%) (*p* = 0.24), possibly due to study underpowering. In a single-centre RCT by Mattiuzzi et al. [23] the incidence of IFI was the same (4%) in both L-AmB (3 mg/kg × 3/week) and fluconazole+itraconazole arms. Finally, in a multi-centre RCT from 1999 [9], there was no ‘proven’ IFI (broadly similar to EORTC/MSG’s ‘probable’ definition [30]) in the L-AmB arm (2 mg/kg × 3/week) vs. three (3.4%) with placebo. This surprisingly low rate of IFI with placebo may stem from the lack of diagnostics, including fungal biomarkers such as galactomannan, available at the time.

Although clinical guidelines exist for diagnostic sampling in high-risk haematology patients, IFI cases may have been missed in this retrospective evaluation. Under a prospective study protocol more instances of clinical deterioration may have been classified as ‘possible’ IFI, and with increased utilisation of bronchoscopy more diagnoses of ‘probable’ IFI might have been made. Inferring a direct link between L-AmB and rates of breakthrough IFI, in particular comparisons between doses and regimens, are limited by historic trends and the absence of a comparator group. Similarly, it is difficult to ascertain whether toxicity observed was directly attributable to L-AmB given the intensive chemotherapy/HSCT conditioning that our patients were also receiving. Moreover, some AEs not recorded in patient notes (e.g., mild infusion reactions) may have been missed.

## 5. Conclusions

To conclude, with an ever-expanding list of triazole drug–drug interactions and additive toxicities to consider, in our large single-centre experience, intermittent high-dose L-AmB prophylaxis represents a safe and effective alternative in high risk haemato-oncology patients. To mitigate the inevitable but reversible common renal toxicities of L-AmB, we recommend stringent monitoring and pre-emptive hydration and electrolyte replacement, especially in the context of HSCT patients on concomitant nephrotoxic agents. As the list of novel immunosuppressive agents associated with IFI risk expands, prospective, adequately powered non-inferiority trials comparing L-AmB to triazoles and novel oral antifungal prophylactic agents are needed to address the ever-present threat that IFIs pose to advances in the treatment of haematological malignancy.

## Figures and Tables

**Table 1 jof-06-00385-t001:** Summary of studies on the use of L-AmB prophylaxis.

Study Type, Year, [Reference]	Population (Total No of Patients)	L-AmB Dosing Schedule (No. of Patients)	Nephrotoxicity, No of Patients (%)	Hypokalaemia No of Patients (%)	Hypomagnasaemia No of Patients (%)	Infusion-related Reaction, No of Patients (%)	Rate of IFI (Timepoint)	Mortality (Timepoint)
Prospective, multi-centre, double-blind, randomised placebo-controlled trial. 1999 [9]	BMT or intensive chemotherapy for haematological malignancy, Adults (161)	2 mg/kg three times per week (74) vs. placebo (87)	Grade 2–4, 9 (12%) vs. 6 (7%)	Grade 4, 1 (1.4%) vs. 0	NR	5 (6%) leading to cessation vs. 1 (1%)	0 proven/probable, 31 (42%) suspected vs. 3 (3.4%) proven/probable, 40 (46%) suspected with placebo * (TU). 15 (20%) had fungal colonisation from at least one body site vs. 35 (40%), *p* < 0.01	11 (15%) vs. 12 (14%) (TU), 1 in each arm attribute to IFI (3–182 days after the end of the study]
Prospective, single-centre, non-blinded, randomised. 2001 [22]	SCT, intensive chemotherapy for haematological malignancy, MDS, VSAA, Children (29)	1 mg/kg three times weekly (16) vs. no systemic antifungal prophylaxis (13)	0	0	0	3/16 (19%) leading to cessation	5/16 (31%) proven/probable vs. 6/13 (46%) (1 year)	0 “by proven IFI” (1 year)
Prospective, single-centre randomised. 2003 [23]	Induction chemotherapy for newly diagnosed AML or high-risk MDS, Adults (139)	3 mg/kg three times per week (72) vs. fluconazole plus itraconazole (67)	Grade 1,2, 14 (20%) vs. 4 (6%)	NR	NR	5 (7%). 10 withdrawn because of drug-related toxicities vs. 5 in F&I arm	3 (4%) proven/probable vs. 3 (4%) (during prophylaxis)	10 (14%) vs. 8 (12%), 1 in each arm attribute to IFI (30 days)
Prospective, single-centre, non-comparative. 2006 [24]	Allo-SCT with GvHD, Adults (21)	7.5 mg/kg once weekly (21) ^‡^	Grade 2–4, 5 (24%) leading to cessation	NR	NR	6 (29%), leading to cessation in 2 (10%)	1 (5%) proven/probable, (2 months)	8 (38%) (median follow up 377 days)
Prospective, singe centre, randomized. 2006 [25]	Intensive chemotherapy for hematological malignancies, Adults (132)	50 mg alt days (75) vs. no systemic antifungal prophylaxis (57)	0 grade 3/4	0 grade 3/4	0 grade 3/4	5 (7%) leading to cessation in 3	5/75 (7%) proven/probable (2 Proven IPA, 3 candidaemia) vs. 20/57 (35%) with no prophylaxis (*p* = 0.001) (during prophylaxis)	4 (5%) vs. 9 (16%), (*p* = 0.13), mortality related to IFI 2 vs. 8 (*p* = 0.07) (median follow up 17 days)
Prospective, multicentre, non-comparative. 2007 [14]	Allo-SCT or chemotherapy for acute leukaemia (AL), Adults (29)	10 mg/kg once weekly (29) ^‡^: 8 SCT ** and 21 AL	Grade 2–4, 4 (14%), lading to cessation in 3 (10%)	2(7%) grade 3	NR	12 (41%) leading to cessation 1 AL and 5 SCT patients. 1 case of anaphylaxis.	4 (14%) proven/ probable (1 SCT, 3 AL). 17 (59%) switched to empirical treatment (during prophylaxis)	4 (2 and 2) (24 weeks)
Prospective, single centre, randomised, pharmacokinetic. 2009 [12]	Allo and Auto-SCT for multiple myeloma, Adults (16)	15 mg/kg once only (6) vs. 7.5 mg/kg once weekly (4) ^‡^ vs. 1 mg/kg daily (6)	Grade 2–4, 3/6 (50%) vs. 1/4 (25%) vs. 1/6 (17%)	Grade 3 15 mg/kg: 1/6 (17%) of patients. Grade 2 2/6 (33%) vs. 1/4 (25%) vs. 1/6 (17%).	6/6 (100%) vs. 4/4 (100%) vs. 5/6 (83%)	1/6 (17%) vs. 2/4 (50%), 1 leading to cessation vs. 2/6 (33%), 1 leading to cessation.	NR	NR
Retrospective, single-centre, comparative. 2009 [26]	Induction chemotherapy for newly diagnosed AML or high-risk MDS, Adults (730)	3 mg/kg three times per week (69) vs. 9 mg/kg once weekly (27) ^‡^ also 6 other arms AmB Lipid Complex, Itra-, flucon plus itra-, vori-, caspo-	NR	NR	NR	Drug related side effects 14% vs. 12%	3 (4%) vs. 2 (7%) (TU)	NR
Retrospective, single-centre, comparative. 2010 [11]	Allo-SCT with GvHD, Adults (125)	7.5 mg/kg once weekly (42) ^‡^ vs. ‘Other antifungal prophylaxis’ (83) of which—59 (72%) fluconazole, 13 (16%) other azole	5 (12%), reversible	NR	NR	NR	4/42 (10%) Proven/Probable plus 2 (5%) Possible. In control arm 13/83 (16%) proven/probable plus 22 (27%) possible (1 year)	14(33%) vs. 35 (42%) (*p* = 0.256). IFI-related mortality 0 vs. 12 (14%) (1 year) (*p* = 0.005) (1 year)
Prospective, single-centre, comparative. 2011 [27]	Intensive chemotherapy for haematological malignancy, VSAA, Children (89)	2.5 mg/kg twice weekly (44) vs. historical control group (45)	7 (16%), with 1 grade 3/4	25/184 (13.6%) grade 3/4, 3/184 grade 4	NR	4/184 (2%) prophylaxis episodes leading to cessation	0 proven/probable vs. 7 (16%) in historical control group (*p* = 0.01) (2 months)	8 (18%) mortality (median follow up 29 months)
Prospective, single centre, non-comparative. 2013 [16]	Induction chemotherapy for AML, Adults, (48)	15 mg/kg once only, followed by second dose after 15 days neutropenia (48) ^‡^	4 (8%), 0 grade 3/4	6 (12.5) grade 3	0	6/53 (11.3%) of infusions cessation in 1/53 (1.9%)	4 (8.3%) proven (4 weeks)	13 (27%) (3 months)
Retrospective, single centre, non-comparative. 2014 [17]	ALL, or solid tumour with prolonged neutropenia, Children (19)	10 mg/kg once weekly (19) ^‡^	1 (5%) leading to cessation	7 (37%) grade 1-3.	2 (10%) any grade	5 (26%) all leading to cessation	1 possible (5%) (during prophylaxis)	1 (5%) (3 months]
Retrospective, single-centre, comparative. 2014 [18]	Allo-SCT with GvHD, Adults (101)	3 mg/kg once weekly (16) vs. echinocandin (12) vs. triazole (73)	0 discontinuation	0 discontinuation	NR	0 discontinuation	0/16 proven/probable vs. 2/12(17%) vs. 1/73 (1%). 3 possible (19%) vs. 0 vs. 3/73 (4%) (*p* = 0.145) (during prophylaxis)	2 (12.5%) neither deemed related to IFI (3 months)
Prospective, single-centre, non-comparative. 2015 [19]	Liver transplant recipients with risk factors for IFI, Adults (76)	10 mg/kg once weekly (76) ^‡^	3 (4%) grade 3	0 (grade 3/4)	NR	6 (8%) leading to discontinuation	3 (4%) proven/probable (2 invasive candidiasis, 1 IPA), 1 possible (during prophylaxis)	20 (26%) all-cause, 0 attributed to IFI (180 days)
Prospective, multi-centre, randomised. 2015 [20]	Induction chemotherapy for haematological malignancy, Adults (52)	1 mg/kg daily (16), 3 mg/kg twice weekly (3), 3 mg/kg three times weekly (18) or 10 mg/kg once weekly (15) ^‡^	NR	NR	NR	AE in 6/16 (37.5%), 2/3 (66.7%), 10/18 (55.6%) and 9/15 (60%) respectively. Serious AE in 1/16 (6.3%), 0/3 (0%), 2/18 (11.1%) and 2/15 (13.3%)	Proven/probable in 1/16 (6.3%), 0/3 (0%), 2/18 (11.1%) and 1/15 (6.7%) respectively (*p* = 0.89) (TU)	5/16 (31.3%), 1/3 (33.3%), 1/18 (5.6%) and 4/15 (26.7%), respectively (*p* = 0.25). 2 IFD-related deaths in the 1 mg/kg daily arm but no other IFD-related deaths. (52 days)
Prospective, multi-centre, double-blind, randomised placebo-controlled trial. 2017 [21]	Induction chemotherapy for ALL, Adults. (355)	5 mg/kg twice weekly (237) vs. placebo (118)	22/237 (9%) creatinine increased but 0 grade 3/4 vs. 0 with placebo	83/237 (35%) of which 13 (5%) grade 3/4 vs. 21/118 (18%) of which 3 (3%) grade 3/4 with placebo	NR	AE leading to study drug discontinuation 63/237(27%) vs. 26/118 (22%)	18/228 (8%) proven/probable vs. 13/111 (12%) in the placebo group (*p* = 0.24). 11/228 (5%) possible vs. 6/111 (5%) in the placebo group (*p* = 0.82) (30 days)	17/237 (7%) vs. 8/118 (7%) for placebo (*p* = 1.00) (30 days)

AE, Adverse event; ALL, acute lymphoblastic leukaemia; Allo, allogenic; BMT, bone marrow transplant; IFI, invasive fungal infection; IPA, invasive pulmonary aspergillosis; L-AmB; Liposomal Amphotericin B; MDS, Myelodysplastic Syndrome; NR, not reported; SCT, stem cell transplant; TU, timepoint unclear; VSAA, Very Severe Aplastic Anaemia. * Proven fungal infection was defined as the microbiological identification of a fungal pathogen associated with clinical or radiological evidence of a disease process likely to be caused by this pathogen. Suspected systemic fungal infection was defined as the presence of fever (37.5 °C) refractory to at least 96 h of therapy with broad-spectrum antibacterial agents or the presence of symptoms or signs consistent with invasive fungal infection, excluding superficial mucocutaneous infection, while on antibacterial therapy ** Enrolment discontinued in the SCT group, as recommended by the independent data review committee, in accordance with the study protocol ^‡^ Intermittent weekly high-dose L-AmB regimens.

**Table 2 jof-06-00385-t002:** Baseline characteristics and rationale for intermittent high-dose L-AmB prophylaxis (*n* = 114).

Characteristic	Frequency	
Age, Median (IQR)	42 (29–57)	
Male, *n* (%)	65	57%
Rationale for antifungal prophylaxis		
Intensive chemotherapy/novel agent (induction/relapse) ^a^	88	77%
HSCT (All)	14	12%
Autograft (pre-engraftment) ^b^	7	6%
Allograft (<100 days post-) ^c^	7	6%
Immunosuppression for GvHD ^d^	10	9%
Prolonged neutropenia (disease related) ^e^	2	2%
Rationale for L-AmB (over triazole)		
Triazole drug-drug interaction	58	51%
*Vincristine*	45	39%
*Myelotarg*	6	5%
*Other* ^f^	7	6%
Abnormal liver function tests	44	39%
Unable to tolerate PO/absorption concerns	6	5%
Cardiomyopathy/risk of QTc prolongation	3	3%
Previous side effects/intolerance to triazole	3	3%
Reason L-AmB prophylaxis discontinued		
Triazole no longer contraindicated	69	61%
Antifungal prophylaxis no longer indicated	16	14%
Death/palliation	7	6%
Infusion reaction	7	6%
Nephrotoxicity	5	4%
Confirmed/suspected IFI	3	3%
Acute allergic reaction	5	4%
Other ^g^	2	2%

^a^ ALL 47, AML 22, NHL 7, HLH 6, MM 2, AA 2, MDS 2. ^b^ NHL 5, MM 2. ^c^ MDS 2, AA 2, AML 1, ALL 1, HL 1. ^d^ ALL 4, AML 2, AA 1, MDS 1, MF 1, HL 1. ^e^ AML 1, AA 1. ^f^ Other triazole drug-drug interactions included: Ritonavir (*n* = 1), Thiotepa (*n* = 2), Rifater/Rifanah (*n* = 2), Ethambutol *n* = 1), Imatinib (*n* = 1), Colchicine (*n* = 1). ^g^ Other reason L-AmB discontinued: Transfer of care (*n* = 1), Noted to have reduced eGFR at baseline (*n* = 1). AA, Aplastic anaemia. ALL, Acute lymphocytic leukaemia. AML, Acute myeloid leukaemia. GvHD, Graft vs. Host Disease. HL, Hodgkin lymphoma. HLH, Hemophagocytic lymphohistiocytosis. HSCT, Hematopoietic stem cell transplantation. IFI, Invasive fungal infection. L-AmB, Liposomal Amphotericin B. MDS, Myelodysplastic syndrome. MM, Multiple myeloma. MF, Myelofibrosis. NHL, Non-Hodgkin lymphoma. PO, Per oral.

**Table 3 jof-06-00385-t003:** Adverse events, breakthrough IFI and mortality (*n* = 114).

L-AmB Prophylaxis Duration		
Course length, median days (range)	24 (7–147)	
Number of infusions, median (range)	3 (1–21)	
Adverse events ^a^		
Acute allergic reaction	5	4%
*Grade 3*	2	2%
*Grade 4*	1	1%
Infusion reaction	7	6%
Nephrotoxicity	65	57%
*Grade 3*	8	7%
Hypokalaemia	75	66%
*Grade 3*	22	19%
*Grade 4*	4	4%
Hypomagnesaemia	75	66%
*Grade 3*	1	1%
Breakthrough IFI	2	2%
*Possible*	1	1%
*Proven/Probable*	1	1%
90 day mortality	10	9%
*With IFI (n = 2)*	2	100%
*Without IFI (n = 112)*	8	7%

^a^ Reported as any grade, followed by grade 3–4 reported separately if present. i.e., For ‘infusion reaction’ all reactions were grade 1, 2. IFI, Invasive fungal infection. L-AmB, Liposomal Amphotericin B.

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
