# Peer review of "Safety and Efficacy of Intermittent High-Dose Liposomal Amphotericin B Antifungal Prophylaxis in Haemato-Oncology: An Eight-Year Single-Centre Experience and Review of the Literature"

_jof, 2020, doi:10.3390/jof6040385_

Round 1
Reviewer 1 Report
The study evaluates the safety, tolerance and efficacy of weekly prophylactic administration of once-weekly high dose (7.5 mg/kg) between Feb 2012 and Jan 2020. It is a retrospective, single-center, observational study of 3rd line prophylactic agents for patients at high risk of IFI in this center. The strength of this manuscript is a large number of patients. Tolerability of L-AMB is what might be expected. Hypokalaemia is the most common grade 3-4 adverse event and grade 3 nephrotoxicity occurs in 7% and reverses in patients surviving to 90 days.
Authors report an IFD rate of 1.8% even if the study is not designed to prove the efficacy of this prophylactic approach.
Questions:
When IFI prophylaxis is started, compared to the first dose of chemotherapy?
What was neutropenic status of patient at base line and how many patients were experienced neutropenia for more than 10 days?
Minor:
In Supplementary Table S1:
Line: Drugs affecting stomach pH e.g. proton pump inhibitors (PPIs) and histamine receptor-2 antagonists (H2 blockers)
Correct: Drugs lowering gastric pH reduce bioavailability of itraconazole capsules (but not oral solution) and posaconazole oral suspension (but not gastro-resistant tablets). PPIs increase Voriconazole exposure and azole CYP3A4 inhibition increases PPI exposure by: Drugs lowering gastric pH reduce bioavailability of itraconazole capsules (but not oral solution) and posaconazole oral suspension (but not gastro-resistant tablets). PPIs increase Voriconazole exposure by CYP2C19 inhibition and azole CYP3A4 inhibition increases PPI exposure.
Author Response
|
REVIEWER 1 |
|
|
Thank you so much for your time spent reviewing our piece, and your helpful and constructive comments which have improved it. |
|
|
Comment |
Response |
|
When IFI prophylaxis is started, compared to the first dose of chemotherapy? |
The following line has been added to methods: “Where antifungal prophylaxis relates to intensive chemotherapy (or a novel agent) it is usually commenced at the time of, or prior to, the first dose.” |
|
What was neutropenic status of patient at base line and how many patients were experienced neutropenia for more than 10 days? |
The following has been added to results:
“Of the 88 patients receiving antifungal prophylaxis in relation to intensive chemotherapy / novel agent, 79 (90%) were severely neutropenic (<0.5 x 109/L) at some point during their L-AmB course. Severe neutropenia lasted over 10 days in 56 (71%) of these courses and occurred a median of 3 days (IQR, 0-7) after L-AmB commenced.” |
|
In Supplementary Table S1: Line: Drugs affecting stomach pH e.g. proton pump inhibitors (PPIs) and histamine receptor-2 antagonists (H2 blockers) Correct: Drugs lowering gastric pH reduce bioavailability of itraconazole capsules (but not oral solution) and posaconazole oral suspension (but not gastro-resistant tablets). PPIs increase Voriconazole exposure and azole CYP3A4 inhibition increases PPI exposure by: Drugs lowering gastric pH reduce bioavailability of itraconazole capsules (but not oral solution) and posaconazole oral suspension (but not gastro-resistant tablets). PPIs increase Voriconazole exposure by CYP2C19 inhibition and azole CYP3A4 inhibition increases PPI exposure. |
Thank you, corrected. |
Reviewer 2 Report
This is an interesting work by Joungs et al regarding the safety and efficacy of intermittent high-dose LAMB antifungal prophylaxis in patients with hematologic malignancies.
Indeed, although azoles have an important role in the prevention of IFD in high-risk patients with hematologic malignancies, we are increasingly facing the hard issue of drug-drug interactions and toxicity.
The current study represents the largest study evaluating the use of intermittent high-dose LAMB administration in the prophylactic setting. Moreover, it summarizes/reviews safety and efficacy data of intermittent LAMB for prophylaxis derived from previous studies, as well as available PK data supporting intermittent use.
The Tables are comprehensive and the article is well-structured. Methods and Results are clear and the Discussion is well-written.
I only suggest a few corrections:
-The first sentence (introduction, lines 39-40) should be rephrased since not all patients undergoing cytotoxic chemotherapy are in need of IFI prophylaxis
-Line 63: undergoing allograft transplantation
-Please review the manuscript once more for minor typo errors (eg. line 85 “who are routinely are cared”, line 264 “there was fewer episodes”)
- Perhaps the Authors should not focus that much to drug-drug interactions between azoles and novel treatments of hematologic malignancies since they have been well covered by other studies.
Author Response
|
REVIEWER 2 |
|
|
Thank you so much for your time spent reviewing our piece, and your helpful and constructive comments which have improved it. |
|
|
Comment |
Response |
|
The first sentence (introduction, lines 39-40) should be rephrased since not all patients undergoing cytotoxic chemotherapy are in need of IFI prophylaxis
|
Word ‘selected’ added:
“Antifungal prophylaxis is recommended for selected patients undergoing cytotoxic chemotherapy and hematopoietic stem-cell transplantation (HSCT) to reduce the risk of invasive fungal infection (IFI) including candidiasis, aspergillosis and mucormycosis. Because of their broad antifungal spectrum, ease of administration, and ability to achieve target serum concentrations, mould-active triazoles (e.g. itraconazole/posaconazole) remain first line agents in high-risk patients such as those with acute myeloid leukaemia/myelodysplastic syndromes (AML/MDS) or undergoing treatment for graft-versus-host disease (GvHD).” |
|
-Line 63: undergoing allograft transplantation
|
Thank you, amended. |
|
-Please review the manuscript once more for minor typo errors (eg. line 85 “who are routinely are cared”, line 264 “there was fewer episodes”)
|
Thank you, these two errors corrected and manuscript reviewed once again. |
|
- Perhaps the Authors should not focus that much to drug-drug interactions between azoles and novel treatments of hematologic malignancies since they have been well covered by other studies. |
Thank you. Whilst we do agree that other studies have discussed this issue we feel that drug-drug interactions between azoles and novel treatments are likely to be an increasing issue in the future. It is for this very reason that we thought it would be timely to share our experience of using weekly L-AmB and would like to highlight this in the text. We note that reviewer 1 appeared to find the Supplementary Table 1 (concerning drug interactions) useful. |